# Application of the Entropy Spectral Method for Streamflow and Flood-Affected Area Forecasting in the Brahmaputra River Basin

**DOI:** 10.3390/e21080722

**Published:** 2019-07-25

**Authors:** Xiaobo Wang, Shaoqiang Wang, Huijuan Cui

**Affiliations:** 1Key Laboratory of Ecosystem Network Observation and Modeling, Chinese Academy of Sciences, Beijing 100101, China; 2Institute of Geographic Sciences and Natural Resources Research, Chinese Academy of Sciences, Beijing 100101, China; 3University of Chinese Academy of Sciences, Beijing 100049, China

**Keywords:** Burg entropy, configurational entropy, streamflow forecasting, flood-affected area, microwave sensors

## Abstract

Reliable streamflow and flood-affected area forecasting is vital for flood control and risk assessment in the Brahmaputra River basin. Based on the satellite remote sensing from four observation sites and ground observation at the Bahadurabad station, the Burg entropy spectral analysis (BESA), the configurational entropy spectral analysis (CESA), maximum likelihood (MLE), ordinary least squares (OLS), and the Yule–Walker (YW) method were developed for the spectral analysis and flood-season streamflow forecasting in the basin. The results indicated that the BESA model had a great advantage in the streamflow forecasting compared with the CESA and other traditional methods. Taking 20% as the allowable error, the forecast passing rate of the BESA model trained by the remote sensing data can reach 93% in flood seasons during 2003–2017, which was significantly higher than that trained by observed streamflow series at the Bahadurabad station. Furthermore, the segmented flood-affected area function with the input of the streamflow forecasted by the BESA model was able to forecast the annual trend of the flood-affected area of rice and tea but needed further improvement in extreme rainfall years. This paper provides a better flood-season streamflow forecasting method for the Brahmaputra River basin, which has the potential to be coupled with hydrological process models to enhance the forecasting accuracy.

## 1. Introduction

The Brahmaputra River basin in Northeastern India annually suffers from large-scale flooding and a large number of agricultural losses owing to the heavy monsoon rainfall and the low-lying terrain [1]. Timely and reliable forecasting of the flood-affected agricultural area is vital for flood control and risk assessment in the Brahmaputra River Basin. Furthermore, it can also provide guidelines for local farmers and policy makers in the utilization of agricultural water resources and adaptive crop planting. The flood-affected agricultural area can be investigated by employing the spatial distribution of the flood-affected area (FAA), the land use, and the topography of the river’s basin [2,3,4]. The flood-affected area can be indirectly estimated from the forecasting of streamflow series. For long-term streamflow prediction, the most frequently used methods to forecast streamflow are time series analyses such as autoregressive (AR) or autoregressive moving average (ARMA) models, which are based on the observed streamflow derived from hydrological stations or satellite remote sensing sites in the flood season [5,6,7]. However, such models suppose that the streamflow time series is stochastic and linear, which limits their application in terms of the strong seasonality that exists in the monthly streamflow series of the Brahmaputra River [8,9]. Entropy spectral analysis has the advantage of extracting periodic and seasonal change information from time series data [10]. Hence, applying entropy spectral theory to time series analysis may provide a reasonable method for streamflow and flood forecasting.

Burg proposed the concept of information entropy in the frequency domain and calculated it by using the power spectral density function of the time series, which has the property of a probability density so that the maximum Burg entropy can obtain the best estimation of the spectral density function of the time series [11]. The method termed Burg entropy spectral analysis (BESA) has been widely used in flood forecasting [12], streamflow forecasting [13], and groundwater level forecasting [14]. Compared with the traditional spectral analysis methods, BESA has the advantage of less assumptions, a higher resolution, and short sequences analysis [15]. Nonetheless, the spectral estimate obtained by the Burg algorithm sometimes displays line splitting and peak shifting. Given that the timing of the monthly streamflow of the Brahmaputra River has presented multi-peak characteristics [16], which is important for forecasting the agricultural flood-affected area, the maximum configurational entropy spectral method (CESA) is likely to be a substitute for the forecasting of multi-peak spectra series.

CESA was initially proposed by Frieden [17] for image recognition. In the field of time series analysis, CESA performs better than BESA in the determination of the spectral density function in the ARMA model and the MA model but has no practical advantage in the AR model [9]. Cui et al. [18] inversely deduced a CESA-based streamflow forecast model by the spectral density estimation formula constructed by Wu [19] and tested CESA using the monthly streamflow data from 19 river sites in the United States. Testing shows that CESA captures streamflow seasonality, satisfactorily forecasts both high and low flows, and shows a higher resolution than BESA.

The above time series method requires observation to train the model and to calculate the parameters, while observations from a few river sites generally do not meet the needs to estimate the flood-affected area of the whole river basin. Actually, on account of the complex geographical, economic, and political factors in the Brahmaputra River basin, it is impractical to obtain continuous streamflow series with full details from hydrological stations from upstream to downstream [20]. In this regard, microwave remote sensing can be used as an alternative source of surface water information for hydrological applications in the Brahmaputra River basin because of its continual, rapid, and dynamic monitoring of the large-scale environment. Spaceborne passive microwave remote sensing such as AMSR-E (Advanced Microwave Scanning Radiometer for Earth observing system) can provide the approximated river streamflow signals using the techniques proposed by Brakenridge et al. [21], and the applicability of assimilating AMSR-E to improve flood prediction in the sparsely gauged Cubango River basin has been previously demonstrated [7]. Hirpa et al. [22] found that the cross-validation regression model derived from the satellite remote sensing series produces the lower streamflow forecast error for high flows compared with low flows, and it has considerably better skill than ARMA model trained by in situ discharge for the Brahmaputra River. Because of the effect of river geometry, topography, and precipitation spatial scale [22], remote sensing streamflow signals may not share the same patterns and neither do their spectra with in situ observations.

Many studies have reported the application of entropy spectral analysis in monthly streamflow and flood forecasting based on hydrological stations, but these methods have been rarely used in spaceborne passive microwave streamflow signals. Therefore, the main objectives of this paper are (1) to compare the forecast performances of BESA and CESA with traditional methods including maximum likelihood (MLE), ordinary least squares (OLS), and the Yule–Walker (YW) method for monthly streamflow forecasting based on the ground observed data and microwave remote sensing data and (2) to simulate the annual agricultural flood-affected area by forecasting the monthly streamflow in the Brahmaputra River Basin in India.

## 2. Methods

The research workflow scheme is shown in Figure 1 and methodology of key processes are reviewed in this section. First, BESA, CESA, MLE, OLS, and YW were applied in parameters estimation of AR models to forecast streamflow in this study. Second, we extracted FAA of rice and tea by spatial analysis of the normalized differences water index (NDWI) and agricultural land use data. Lastly, a FAA function was established based on the sub-basin and whole-basin scale analysis and streamflow series forecasted by an optimal model was used to predict the FAA in the Brahmaputra River Basin.

### 2.1. Streamflow Forecasting Based on Maximum Burg Entropy Spectrum Analysis

As introduced by Burg [11] and Singh [23], maximum Burg entropy spectrum analysis regarded the frequency *f* of time series as a random variable and defined Burg entropy as
(1)$$H_{B}(f)=-\int_{-W}^{W}\ln[P(f)]df$$
where *W* = ½ × Δ*t* is the Nyquist fold-over frequency and Δ*t* is the sampling period. *P*(*f*) is a power spectral density function and it is also a probability density function.

Maximum entropy is obtained with constraints formed from the power spectral density function. For a stationary random sequence, the autocorrelation function *ρ*(*n*) and the power spectral density function *P*(*f*) are Fourier transforms of each other. The constraints of discrete time series can be expressed as
(2)$$\rho (n)=\int_{-W}^{W}P(f)e^{i2\pi fn\Delta t}df,-N\leq n\leq N$$
where *i* is an imaginary number, and *n* is the time-lag of the autocorrelation function and also represents the *n*-th harmonic wave of the discrete Fourier transform.

When *n* = 0, Equation (2) reduces to
(3)$$1=\int_{-W}^{W}P(f)df$$

In order to solve the power spectral density function corresponding to the maximum Burg entropy, the Lagrange multiplier method is used to construct the Lagrange function. The Lagrange function based on the Burg entropy is expressed as
(4)$$L_{B}(f)=-\int_{-W}^{W}\ln[P(f)]df-\sum_{n=-N}^{N}\lambda_{n}\left[\int_{-W}^{W}P(f)e^{i2\pi fn\Delta t}df-\rho (n)\right]$$
where *λ_n_* is the Lagrange multiplier.

The partial differential of the Lagrange function *L_B_*(*f*) for *P*(*f*) is calculated and made equal to 0 so that the power spectral density function *P_B_*(*f*) corresponding to the maximum Burg entropy is obtained:(5)$$P_{B}(f)=-\frac{1}{\sum_{n=N}^{N}\lambda_{n}\exp(-i2\pi fn\Delta t)}$$

The form of power spectral density is equivalent to that of an autoregressive model when *P_B_*(*f*) is transformed as
(6)$$P_{B}(f)=-\frac{\sigma_{m}^{2}}{{\left[1+\sum_{n=1}^{m}a_{m}(n)\exp(-i2\pi fn\Delta t)\right]}^{2}}$$

In the formula, *m* is the order of the model and $\sigma_{m}^{2}$ is the variance of the *m*-order model. If the mean value of the input time series is 0, $\sigma_{m}^{2}$ is equal to the output power.

Burg proposed a recursive method to solve the autoregressive model by using the reflection coefficient. The estimation formula of the *i*-th parameter *a_k_*(*i*) of the *k*-order autoregressive model is
(7)$$a_{k}(i)=\{\begin{array}{ll} a_{k-1}(i)+k_{k}a_{k-1}(k-i); & i=1,2,\ldots ,k-1\\ k_{k} & ;i=k \end{array}$$

According to the objective function of minimum power of forward prediction error and backward prediction error, the reflection coefficient *k_k_* of the model can be derived as follows:(8)$$k_{k}=\frac{-2\sum_{i=k}^{N-1}e_{k-1}^{f}(i)e_{k-1}^{b}(i-1)}{\sum_{i=k}^{N-1}{\left|e_{k-1}^{f}(i)\right|}^{2}+\sum_{i=k}^{N-1}{\left|e_{k-1}^{b}(i-1)\right|}^{2}}$$
(9)$$e_{k}^{f}(i)=e_{k-1}^{f}(i)+k_{k}e_{k-1}^{b}(i-1)$$
(10)$$e_{k}^{b}(i)=e_{k-1}^{b}(i-1)+k_{k}e_{k-1}^{f}(i)$$

The best order *m* is determined by Akaike information criterion (AIC) criteria, and the power spectral density function based on BESA can be obtained by substituting {*a_m_*(1), *a_m_*(2), …, *a_m_*(*m*)} into Equation (6). The *m*-order autoregressive model becomes
(11)$$\rho (n)=\sum_{k=1}^{m}a_{m}(k)\rho (n-k)$$

Therefore, the autoregressive model can be applied to monthly streamflow forecasting as below:(12)$$x(t)=\sum_{k=1}^{m}a_{m}(k)x(t-k),t>N$$

### 2.2. Streamflow Forecasting Based on Configurational Entropy Spectrum Analysis

Configurational entropy and Shannon entropy are similar in definition [24]. As with Burg entropy, configurational entropy takes the frequency of streamflow time series as the independent variable:(13)$$H_{C}(f)=-\int_{-W}^{W}P(f)\ln[P(f)]df$$
where *W* = ½ × Δ*t* is the Nyquist fold-over frequency and Δ*t* is the sampling period.

The Lagrange multiplier method is also used to construct the Lagrange function based on the configurational entropy:(14)$$L_{C}(f)=-\int_{-W}^{W}P(f)\ln[P(f)]df-\sum_{n=-N}^{N}\lambda_{n}\left[\int_{-W}^{W}P(f)e^{i2\pi fn\Delta t}df-\rho (n)\right]$$
where *λ_n_* is the Lagrange multiplier.

To obtain the spectral density maximizing the entropy, the partial differential of the Lagrange function *L_C_*(*f*) for *P*(*f*) is calculated and made equal to 0 so that the power spectral density function *P_C_*(*f*) corresponding to the configurational entropy is obtained:(15)$$P_{C}(f)=\exp\left(-1-\sum_{n=-N}^{N}\lambda_{n}e^{i2\pi fn\Delta t}\right)$$

The form of the spectral density function of maximum configurational entropy is an exponential function, which is different from the form of the spectral density function of the maximum Burg entropy. Wu [19] applied cepstrum analysis to the solution of the Lagrange multiplier and extension of the autocorrelation function and proposed the explicit solution of the spectral density function based on the maximum configurational entropy. The inverse Fourier transform of the logarithmic solution of Equation (14) can be obtained as
(16)$$\int_{-W}^{W}\{1+\ln[P(f)]\}\exp(i2\pi k\Delta t)df=\int_{-W}^{W}\left[-\sum_{n=-N}^{N}\lambda_{n}\exp(i2\pi fn\Delta t)\right]\exp(i2\pi fk\Delta t)df$$
where *k* is the *k*-th harmonic of the inverse discrete Fourier transform. Defining
(17)$$e(k)=\int_{-W}^{W}\ln[P(f)]\exp(i2\pi fk\Delta t)df$$

The Equation (18) can be simplified as
(18)$$\delta_{k}+e(k)=-\sum_{n=-N}^{N}\lambda_{n}\delta_{k-n}$$
where *δ_k_* is a unit pulse function.

For *k* with different values, a set of equations for solving Lagrange multipliers can be obtained:(19)$$\begin{array}{c} \lambda_{0}=-e(0)-1\\ \lambda_{1}=-e(1)\\ \vdots \\ \lambda_{N}=-e(N) \end{array}$$

In order to solve the spectral density function conveniently, Nadeu [25] proposed a simple method for cepstrum *e*(*n*) estimation. Its principle is to minimize the difference between the estimated spectral density and prior information. When −*N*
*≤ n*
*≤ N*, the cepstrum formula is expressed as
(20)$$e(n)=\{\begin{array}{c} 2\left[\rho (n)-\sum_{k=1}^{n-1}\frac{k}{n}e(k)\rho (n-k)\right];n>0\\ 0;n\leq 0 \end{array}$$

The autocorrelation coefficient is deduced from cepstrum:(21)$$\rho (n)=\frac{e(n)}{2}+\sum_{k=1}^{n-1}\frac{k}{n}e(k)\rho (n-k)$$

If the order of an autoregressive model is determined to be *m*, the autocorrelation coefficient *ρ*(*m*) is
(22)$$\rho (m)=\sum_{k=1}^{m}a_{m}(k)\rho (m-k),\;m\leq N$$
(23)$$a_{m}(k)=\frac{k}{m}e(k),\;k<m$$

The autoregressive coefficient is extended to forecast streamflow:(24)$$\rho (t)=\sum_{k=1}^{m}a_{m}(k)\rho (t-k),\;t>T$$
where *T* is the total length of streamflow series. Therefore, the forecast model of streamflow *x*(*t*) can be expressed as
(25)$$x(t)=\sum_{k=1}^{m}a_{k}x(t-k),\;t>T$$

### 2.3. Streamflow Forecasting Based on MLE, OLS, and YW

Principles of parameter estimation methods including MLE, OLS, and YW are briefly reviewed in this sub-section. MLE is a method to estimate the parameter *θ* by maximizing the likelihood function *L*(*θ*;*x*), so that the observed data is most probable under the assumed statistical model. To maximize *L*(*θ*;*x*) with respect to *θ*:(1) first calculate the derivative of *L*(*θ*;*x*) with respect to *θ*, (2) set the derivative equal to zero, and (3) solve the resulting equation for *θ*. Contrary to MLE, OLS chooses the parameter *θ* of a set of explanatory variables by minimizing the sum of the squares of the differences between the observed streamflow in the given dataset and those predicted by the linear function.

The Yule–Walker estimation is based on the Yule–Walker equation of the AR model [26,27,28]. The specific steps are as follows:(26)$$\rho (m)=\sum_{k=1}^{p}a(k)\rho (m-k)+\sigma_{e}^{2}\delta_{m,0}$$
where *m* = 1, 2, …, *p*, yielding *p* = 1 equations and *a*(*k*) are parameters of explanatory variables. Here $\sigma_{e}^{}$ is the standard deviation of the input noise process, and $\delta_{m,0}$ is the Kronecker delta function.

Because the last part of an individual equation is non-zero only if *m* = 0, the set of equations can be solved by representing the equations for *m* > 0 in matrix form, thus getting the equation
(27)$$\left[\begin{array}{l} \begin{array}{c} \rho (1)\\ \rho (2)\\ \rho (3)\\ \vdots \end{array}\\ \rho (p) \end{array}\right]=\left[\begin{array}{cccc} \rho (0) & \rho (-1) & \rho (-2) & \ldots \\ \rho (1) & \rho (0) & \rho (-1) & \ldots \\ \rho (2) & \rho (1) & \rho (0) & \ldots \\ \vdots & \vdots & \vdots & \ddots \\ \rho (p-1) & \rho (p-2) & \rho (p-3) & \cdots \end{array}\right]\left[\begin{array}{c} a(1)\\ a(2)\\ a(3)\\ \vdots \\ a(p) \end{array}\right]$$
which can be solved for all {*a*(*m*); *m* = 1, 2, …, *p*}. The remaining equation for *m* = 0 is
(28)$$\rho (0)=\sum_{k=1}^{p}a(k)\rho (-k)+\sigma_{e}^{2}$$
in which, once {*a*(*m*); *m* = 1, 2, …, *p*} are known, can be solved for $\sigma_{e}^{2}$.

### 2.4. Determination of the Order of the Autoregressive Model

This paper used the Akaike information criterion (AIC) to determine the order *m* [29]:(29)$$\mathrm{AIC}=N\ln{\hat{\sigma }}^{2}+m$$
where *N* is the total length of the training series and ${\hat{\sigma }}^{2}$ is the variance of the residual of the model simulation.

### 2.5. Flood-Affected Area Estimation

This paper estimated the flood-affected area by subtracting the spatial distribution of flood inundated pixels in the dry season from that in the flood season, as shown in Equation (27).
*S_faa_ = S_flood_ − S_dry_*(30)

Flood-inundated pixels were identified by the normalized differences water index (NDWI) [30] extracted from remote sensing data: (31)$$NDWI=\frac{G-NIR}{G+NIR}$$
where *G* and *NIR* are the green light and near-infrared reflectance of ground objects, respectively.

The flood-affected area of rice and tea plantations in the basin was obtained by spatial overlay analysis of the flood-affected area and harvest area of rice and tea in the flood season, and it was estimated by single variable regression between the flood-affected area and the flood-season streamflow based on the microwave remote sensing data.

Evaluation indicators used for the forecast accuracy of streamflow and the flood-affected area in this paper were the Pearson correlation coefficient (*R*), root mean square error (*RMSE*), and Nash–Sutcliffe efficiency coefficient (*NSE*), which can be expressed as
(32)$$R=\frac{\sum_{t=1}^{n}(x(t)-\bar{x})(f(t)-\bar{f})}{{\left[\sum_{t=1}^{n}(x(t)-\bar{x}{)}^{2}\sum_{t=1}^{n}(f(t)-\bar{f}{)}^{2}\right]}^{0.5}}$$
(33)$$RMSE=\sqrt{\frac{\sum_{t=1}^{n}{\left[x(t)-f(t)\right]}^{2}}{n-1}}$$
(34)$$NSE=1-\frac{\sum_{t=1}^{n}{\left|x(t)-f(t)\right|}^{2}}{\sum_{t=1}^{n}{\left|x(t)-\bar{x}\right|}^{2}}$$
where *x* represents the average value of observed streamflow *x*(*t*), $\bar{f}$ represents the average value of the forecasted streamflow *f*(*t*), and *n* is the forecasting period.

## 3. Application

### 3.1. Data Sources and Preprocessing

The Brahmaputra River, which originates from the Yarlung–Zangbo River in China, has a total length of 2880 km, and 34% of the basin area is located in India. The climate in the basin can be divided into a dry season (November–April) and a monsoon season (May–October), and precipitation in the monsoon season accounts for more than 85% of the total annual precipitation. The flood season of the Brahmaputra River is between July and October. Rice and tea are the main food crops and cash crops in the Brahmaputra Basin. According to the MAPSPAM2010 dataset, the physical area of rice and tea plantations are 2.33 and 0.38 million hectares respectively, and most plantation areas are located in the Brahmaputra Valley in the Assam state (Figure 2).

In this paper, the spatial distribution of the rice and tea harvest area in Northeast India was obtained from the MAPSPAM2010 database and annual flood-inundated areas during 2001–2017 in the basin were extracted from the Moderate Resolution Imaging Spectroradiometer (MODIS) data. Ground observed streamflow data from the Bahadurabad station (located 25.15° N/89.70° E) in Bangladesh, which controls the whole area of the Brahmaputra River basin in India, and satellite streamflow measurement data from four sites along the river (located 25.51° N/89.78° E, 26.32° N/92.03° E, 27.13° N/94.55° E, and 27.76° N/95.54° E, respectively) were selected to compare the performance of different models. Satellite streamflow data was derived from the River and Reservoir Watch Version 3.5 dataset, which is based on the Advanced Microwave Scanning Radiometer - Earth Observing System (AMSR-E), the Tropical Rainfall Measuring Mission (TRMM), Advanced Microwave Scanning Radiometer-2 (AMSR-2), and Global Precipitation Measurement (GPM) sensors data [31]. 

Streamflow data from the Bahadurabad station and remote sensing were processed on the monthly scale and raster datasets of the harvest area were interpolated with a spatial resolution of 250 m. Given that the input data of autocorrelation models should be a standardized stationary random series, the streamflow series were standardized and transformed using the Box–Cox method. Locations of the Bahadurabad station and satellite river discharge measurement sites are shown in Figure 2. Input data types and sources are shown in Table 1.

### 3.2. Historical Streamflow and Agricultural Flood-Affected Area in the Brahmaputra River Basin

In order to determine the streamflow indicator suitable for forecasting the agricultural flood-affected area in the Brahmaputra River basin, observed historical annual maximum monthly streamflow, maximum daily streamflow, maximum seven-day streamflow, total flood streamflow, annual streamflow, and mean streamflow in the flood season (July–October) from 2001 to 2017 were analyzed. The result showed that the mean streamflow in the flood season has the strongest correlation with the agricultural flood-affected area. 

Inter-annual patterns of mean streamflow in the flood season at different sites and flood-affected areas (FAA) in the Brahmaputra basin are shown in Figure 3. Streamflow in the flood season at five stations during 2001–2017 increased progressively from upstream sites to downstream sites, which ranged from 4411 m^3^/s at Satellite-4 to 36,797 m^3^/s at Bahadurabad, and the average flood-season streamflow of the four satellite sites was 14,782 m^3^/s. According to MODIS data, the maximum annual flood-affected area of rice (FAA_rice_) and tea (FAA_tea_) during 2001~2017 in Northeast India occurred in 2004, which was 115,685 ha and 10,074 ha respectively.

We divided the Brahmaputra basin in Northeast India into four sub-basins based on the locations of streamflow observation sites to calculate the correlation between FAA and mainstream flow (Figure 4). Owing to the impact of both local and upstream runoff to mainstream flood, the observation site satellite-1 controls the streamflow of sub-basins I + II + III + IV, satellite-2 controls sub-basins II + III + IV, satellite-3 controls sub-basins III + IV, and satellite-4 controls the sub-basin IV. Pearson correlation coefficients between the flood-season streamflow and FAA at the sub-basin scale are shown in Table 2 and at the whole-basin scale are shown in Table 3. At the sub-basin scale, the streamflow from satellite-1 and satellite-3 both has a significant correlation with FAA_rice_ (0.667 and 0.691, *p* < 0.01) and FAA_tea_ (0.671 and 0.690, *p* < 0.01) in corresponding sub-regions_._ At the whole-basin scale, the average streamflow of the four satellite sites has a higher correlation coefficient (0.798 with FAA_rice_ and 0.747 with FAA_tea_, *p* < 0.01) than the individual satellite observation site. Compared with the satellite multi-site observation along the river, the streamflow at Bahadurabad shows less correlation with the annual flood-affected area (0.494 with FAA_rice_ and 0.486 with FAA_tea_, *p* < 0.05). Beyond that, determining the coefficient of multivariate regression between FAA and four satellite-derived streamflow series based on sub-basin analysis is lower than that between FAA and mean streamflow series based on the whole-basin analysis. Thus, we chose mean streamflow series of four satellite sites rather than individual sites to test entropy spectral models and forecast FAA in the Brahmaputra basin.

### 3.3. Monthly Streamflow Forecasting

In order to examine whether the entropy spectral method will improve the forecasting precision of flood-season streamflow in the Brahmaputra River basin, we used BESA, CESA, MLE, OLS, and YW methods to forecast the monthly streamflow for selecting the optimal model and compared the model training effects of remote sensing data and hydrological observation series. The average streamflow of the four satellite sites along the Brahmaputra River and monthly streamflow at Bahadurabad hydrological station were chosen as the training data of AR models and the best order was selected under Akaike information criterion for all forecasting experiments in the paper.

Before calibrating the parameters of the AR model, the optimal training period was determined. In this paper, we selected the observed streamflow data from the length of 36 months to 120 months as a training period to evaluate the influence of the training period on the forecasted results. According to the calculation results, when the training period is less than 60 months, the NSE of the forecasting models using BESA/CESA/traditional methods is less than 0.72, and the NSE of models tends to be stable with the increase of training period (Figure 5). Therefore, the streamflow of 12 months per year are forecasted by the previous 60-month period in AR models with feedback in this paper. The forecast results of mean monthly streamflow at the Brahmaputra River based on microwave remote sensing series data during 2003–2017 is shown in Figure 6 and Table 4. Streamflow at Bahadurabad forecasted using hydrological station series data during 2003–2017 is shown in Figure 7 and Table 5.

At the monthly scale from January to December, we found that the rank of forecast accuracy with the evaluation criteria of *R*, RMSE, and NSE was in the order BESA > traditional methods (MLE, OLS, and YW) > CESA for satellite sites and the Bahadurabad station by comparing the monthly streamflow from the five models. Furthermore, the streamflow forecasting accuracy for Bahadurabad station by the five models was higher than the upstream four satellite observation sites with the evaluation criteria of *R* and NSE. The reason for this is that streamflow signals derived from remote sensing were disturbed by external factors such as irrigation in the dry season, which made remote sensing signals show weaker autocorrelation than the streamflow measured at the hydrological station.

At the flood-season scale, on the other hand, it can be discovered that the models trained with Bahadurabad station data overestimated the flood-season streamflow in August and September of 2003 and 2005, resulting in more outliers and a lower accuracy than the models trained by remote sensing data (Figure 8). Taking 20% of the measured streamflow as the allowable error, the average passing rate of the flood-season streamflow forecasting was 65% at Bahadurabad station and was 76% at the satellite sites. Among all the methods, the BESA method had the highest flood-season streamflow forecast passing rate, which reached 73% and 93% when using the Bahadurabad station data and remote sensing data training, respectively. The forecast passing rate of CESA, MLE, OLS, and YW methods was 60%–67% at the Bahadurabad station and 60%–80% at the satellite sites.

In summary, the monthly streamflow forecast accuracy of the BESA method is significantly higher than the traditional methods, regardless of whether it is on a monthly scale or flood-season scale. However, there is no significant difference between the CESA and traditional methods in the forecast accuracy of streamflow at the Brahmaputra River. At the monthly scale from January to December, the models trained by the Bahadurabad station perform better than those trained by microwave remote sensing series. Nevertheless, at the flood-season scale, the models trained by the Bahadurabad station perform worse than those trained by remote sensing series. Given that remote sensing series of streamflow in flood season has a stronger correlation with the flood-affected area, we selected forecasting results from the BESA model based on the remote sensing observation sites during the flood season to forecast the agricultural flood-affected area in the Brahmaputra River basin.

### 3.4. Flood-Affected Area Functions and Forecasting

In order to convert the forecasted flood-season streamflow into the agricultural FAA, regression functions between monthly mean streamflow in flood seasons and the flood-affected area of rice and tea were established based on remote sensing data from 2001 to 2017. The calibration period is 2001–2007 and the validation period is 2008–2017.

We first tested three forms of the flood-affected area functions according to a previous study: linear, exponential, and logistic [1]. The results showed that the *R*^2^ value of the fitted linear function for the flood-affected area of rice and tea is 0.718 and 0.633 respectively; the fitted exponential curve is very close to the fitted logistic function curve, hence their *R*^2^ values for the FAA of rice and tea are 0.773 and 0.647, respectively. Second, considering that there is an abrupt change point in the variation curve of the agricultural flood-affected area with increased streamflow, the segmented function form was also selected to fit the agricultural flood-affected area in the Brahmaputra River basin. The fitted segmented equations for the flood-affected area of rice and tea are shown in Figure 9, where *X* is the mean streamflow in the flood season and FAA_rice_ and FAA_tea_ refer to the flood-affected area of rice and tea, respectively. From Figure 5, we can see that when the average streamflow in the flood season was lower than 11,218 m^3^/s, FAA_rice_ did not change with the streamflow. When the flood-season streamflow was higher than 11,218 m^3^/s, FAA_rice_ increased rapidly with the increase of streamflow. The FAA_tea_ curve was similar to the FAA_rice_ curve, but it had a higher abrupt change point with the flood-season streamflow. The *R*^2^ values of the fitted segmented functions are 0.819 and 0.824, respectively, significantly higher than the linear, exponential, and logistic functions. Therefore, we used the segmented functions to forecast the flood-affected area of rice and tea in the Brahmaputra River basin during 2008–2017.

The validation results showed a significant linear correlation between the forecasted and observed FAA (R = 0.60, *p* < 0.01) during 2008–2017, which indicated that the FAA function can basically estimate the annual trend of the FAA of rice and tea based on the flood-season streamflow. The absolute relative error was 20.12% of forecast FAA_rice_ and 26.45% of forecast FAA_tea_. As a whole, the regression function tends to overestimate the FAA in a dry year and underestimate the FAA in a flood year, especially for tea, which needs further improvement for forecasting the FAA in extreme rainfall years.

## 4. Discussion

Previous studies have manifested that BESA and CESA can both accurately forecast monthly streamflow, and CESA showed a better performance than BESA when fitting streamflow at Iowa River [10], Greenbrier River, Upper Colorado River, and Green River in the United States [18] and rivers in Northwest China [9]. Nevertheless, we found that BESA performed better than CESA in the Brahmaputra River basin. There are three reasons for the results. First, the streamflow series at the Brahmaputra River during 2001–2017 exhibited a strong autocorrelation (R was 0.71 at the 12th lag at the Bahadurabad station) and a previous study discovered that BESA had better forecast results for the streamflow data, with a strong autocorrelation coefficient of more than 0.60 at the lag of 12 months [8]. Second, the order of the CESA model (*m* = 20) determined by AIC was larger than BESA and other traditional models (13 ≤ *m* ≤ 16), which will introduce more uncertainties of streamflow signal and induce over-fit of the CESA model. Third, the temporal distribution of the streamflow at the Brahmaputra River showed bi-modal characteristics at a monthly scale from 2010 to 2017. The model results support the view that the BESA model can more accurately forecast the bi-modal values of the flood season for the lead time [8]. In addition, CESA can be derived with spectral power (CESAS) or frequency (CESAF) as a random variable and it was found that CESAS performed very similarly with BESA in Minnesota River, Upper Mississippi River, Des Moines River, and Illinois River [13]. Performance differences between CESAS and CESAF in Brahmaputra River is not discussed in this paper and need a further study in future.

In this study, the relative coarse spatial resolution (250 m) but temporally dense time series (8-day composition) of inundation information derived from the MODIS data allowed for the revelation of inter-annual FAA dynamics and overall inundation patterns for large areas. Nevertheless, discrepancies between the MODIS derived and the actual inundation areas exist in the boundary regions of the water bodies due to the mixed pixel effects. Kuenzer et al. found that the accuracy of inundation patterns derived from the MODIS data ranged between 79% and 99% in the Ganges–Brahmaputra Delta [35], which may induce errors of flood area estimation. Moreover, the length of the accessible MODIS data is less than 20 years, which maybe not enough for regression analysis. Additionally, the fitted flood-affected area function in the paper can hardly represent the whole basin in the context of climate change and land cover change. Strong increasing trends were predicted for the total water yield and streamflow at the Brahmaputra River basin, indicating the possibility of the sudden change of the streamflow temporal pattern and exacerbation of flooding potential in the future [32,36], while regression models trained with historical observed data are not able to simulate these variations. Therefore, the models developed in the paper are suitable for medium- or short-term forecasts and should be coupled with hydrological process models to prolong the forecast period.

## 5. Conclusions

From the perspective of the river basin planning and management, a reliable streamflow forecast is critical for estimating the agricultural loss due to floods. Using remote sensing data from four observation sites and hydrological station data, we developed the BESA, CESA, and other traditional models for the spectral analysis and flood-season streamflow forecasting of the Brahmaputra River basin. The results indicated that forecast models trained by microwave remote sensing series performed better in the flood season than those trained by hydrological station data at Bahadurabad. The streamflow forecast accuracy of the BESA model was significantly higher than the traditional methods, while the CESA model did not show any advantage in terms of the forecast accuracy of streamflow at the Brahmaputra River, regardless of whether a monthly scale or flood-season scale was being considered. Taking 20% of the measured flood-season streamflow as the allowable error, the forecast passing rate of the BESA model trained by the remote sensing data reached 93% during 2003–2017, which met the basic needs of flood control and risk assessment in the Brahmaputra River basin.

Moreover, the segmented flood-affected area (FAA) function and the streamflow forecast output from the BESA model were utilized to forecast the FAA of rice and tea in the basin during 2008–2017. The prediction results showed that the FAA function was able to forecast the annual trend of the flood-affected area of rice and tea and the absolute relative error of the forecast flood-affected area of rice and tea was 20.12% and 26.45%, respectively. In future research, the streamflow forecast model trained by the microwave remote sensing series with the BESA method can be coupled with hydrological process models to prolong the forecast period and enhance the forecast accuracy of the agricultural loss due to floods.

## Figures and Tables

**Figure 1 entropy-21-00722-f001:**
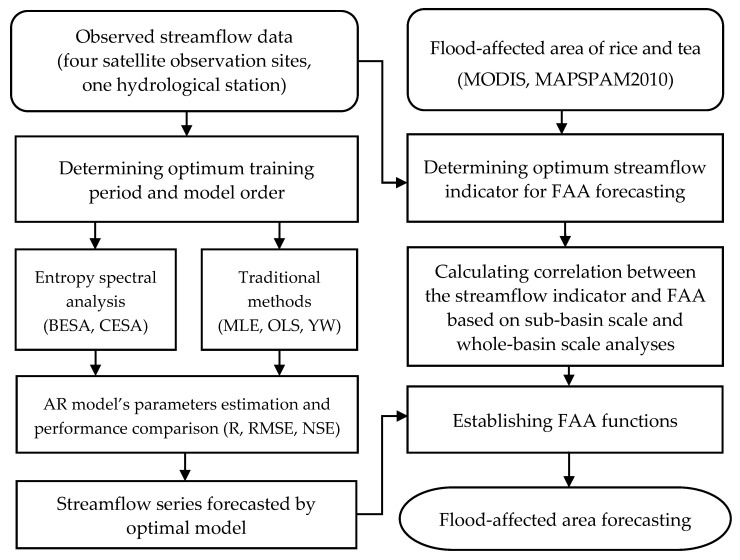
The research workflow scheme of streamflow and flood-affected area forecasting.

**Figure 2 entropy-21-00722-f002:**
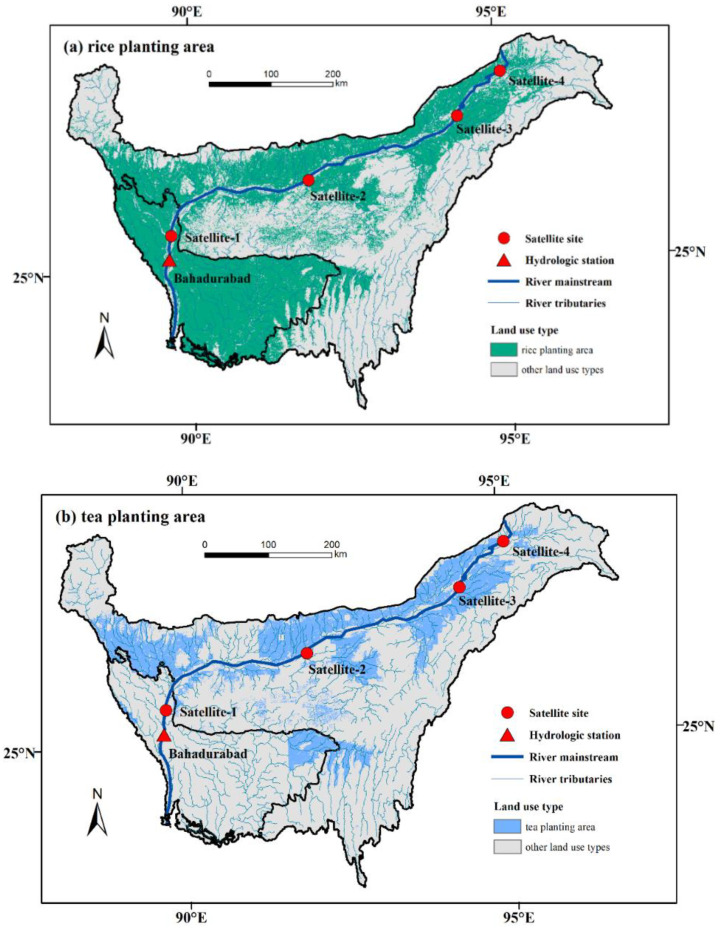
Locations of the hydrological station and satellite river discharge measurement sites in the Brahmaputra River basin. (**a**) Rice planting area; (**b**) tea planting area.

**Figure 3 entropy-21-00722-f003:**
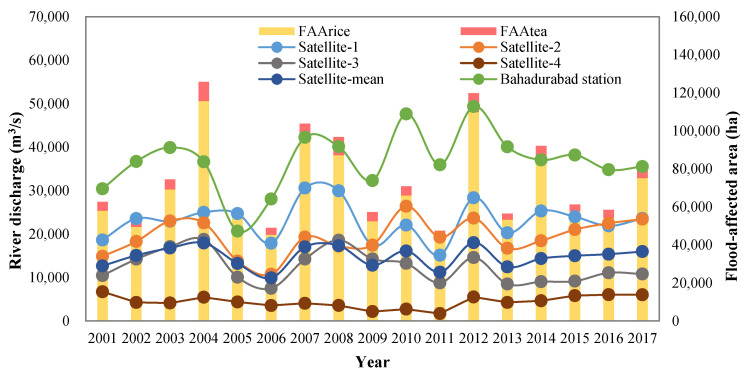
Mean streamflow in the flood season at different sites and flood-affected areas in the Brahmaputra basin during 2001–2017.

**Figure 4 entropy-21-00722-f004:**
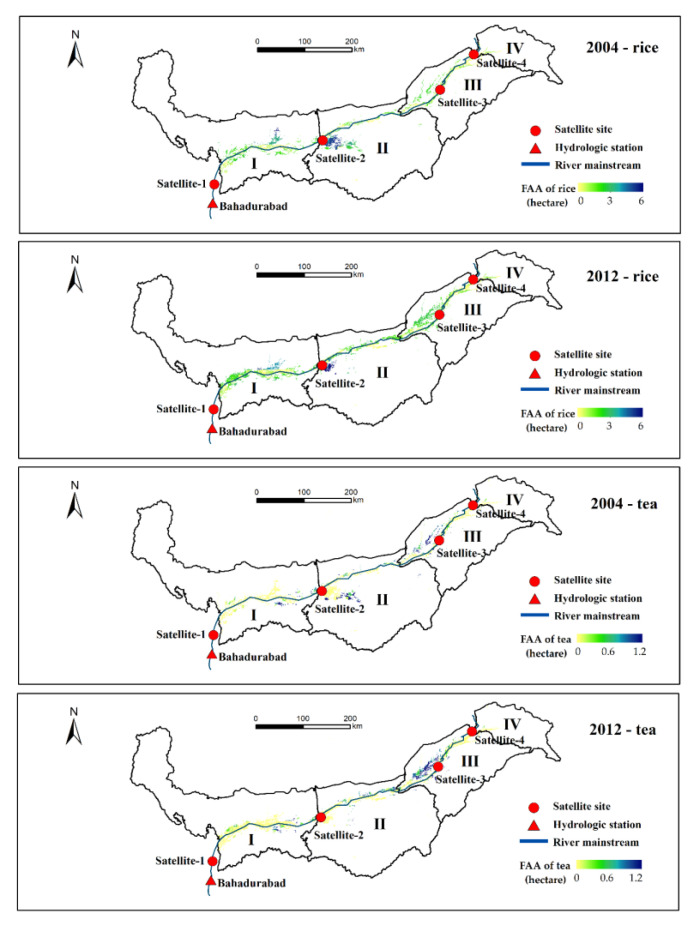
Cross-sections of spatial distribution pattern of the flood-affected area of rice (FAA_rice_) and flood-affected area of tea (FAA_tea_) in 2004 and 2012.

**Figure 5 entropy-21-00722-f005:**
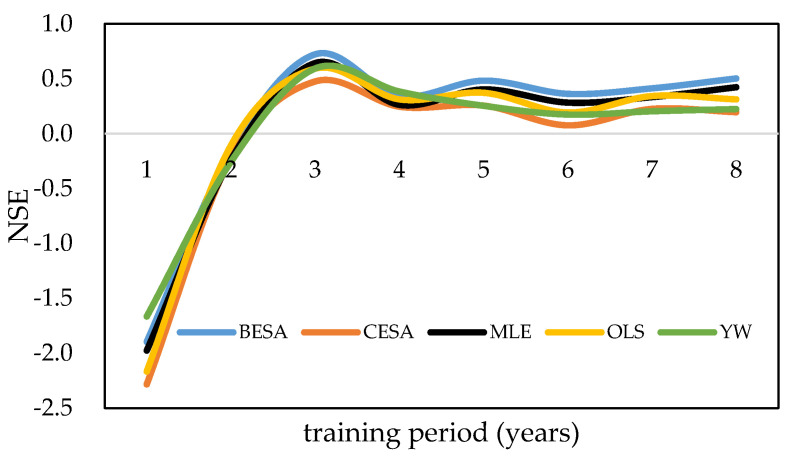
Nash–Sutcliffe efficiency coefficient (NSE) values of models trained by different training periods.

**Figure 6 entropy-21-00722-f006:**
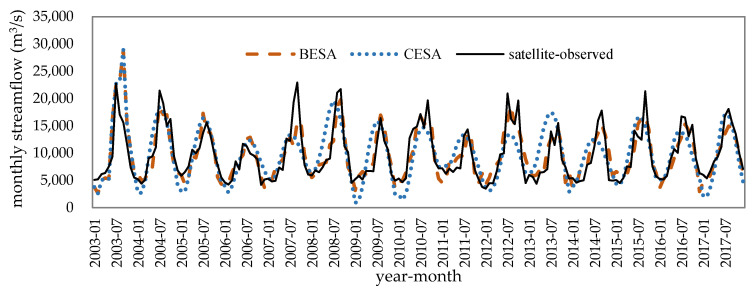
Observed and forecasted streamflow based on the remote sensing series data.

**Figure 7 entropy-21-00722-f007:**
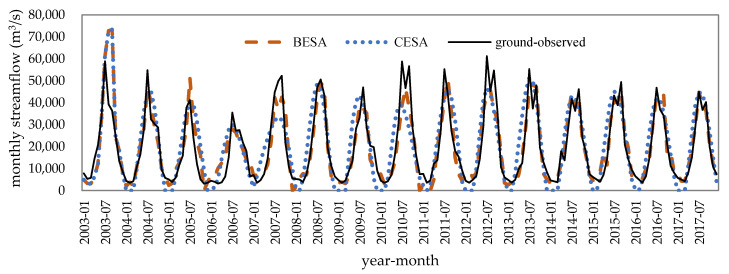
Observed and forecasted streamflow based on the Bahadurabad station data.

**Figure 8 entropy-21-00722-f008:**
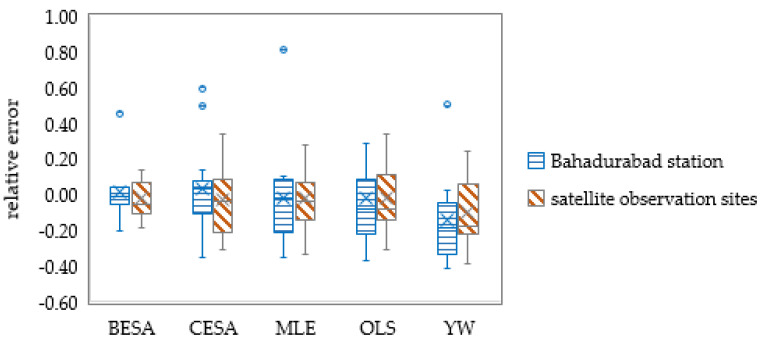
Relative errors of streamflow forecasting in the flood season (blue dots are outliers of forecast results).

**Figure 9 entropy-21-00722-f009:**
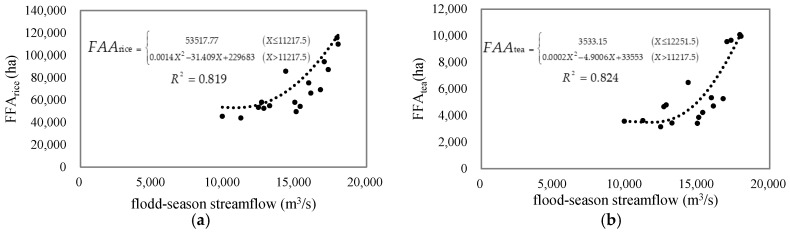
Flood-affected area function curves with observed flood-season streamflow: (**a**) Observed flood-affected area of rice and the segmented function of FAA_rice_; (**b**) observed flood-affected area of tea and the segmented function of FAA_tea_.

**Table 1 entropy-21-00722-t001:** Basic information of datasets used in the paper.

	Data Type	Temporal/Spatial Scale	Data Source
Ground observed streamflow data	Time series	Year 2001–2017, monthly	BWDB [16,32,33]
Satellite streamflow measurement data	Time series	Year 2001–2017, monthly (from daily)	River and Reservoir Watch Version 3.5 [31]
Harvest area of rice and tea	Raster dataset	Year 2010, 250 m (interpolated from 10 km)	MAPSPAM2010 [34]
Flood-inundated Area	Raster dataset	Year 2001–2017, 250 m	MODIS

**Table 2 entropy-21-00722-t002:** Pearson correlation coefficient table of flood-season streamflow and flood-affected area at the sub-basin scale.

Flood-Season Streamflow and Flood-Affected Area		Satellite-1 and Sub-Basin I + II + III + IV	Satellite-2 and Sub-Basin II + III + IV	Satellite-3 and Sub-Basin III + IV	Satellite-4 and Sub-Basin IV
FAA_rice_	Correlation Coefficient	0.667	0.404	0.691	0.385
	Significance	0.003	0.097	0.001	0.115
FAA_tea_	Correlation Coefficient	0.671	0.267	0.690	0.518
	Significance	0.002	0.285	0.001	0.028

**Table 3 entropy-21-00722-t003:** Pearson correlation coefficient table of flood-season streamflow and flood-affected area at the whole-basin scale.

Flood-Affected Area		Satellite-1	Satellite-2	Satellite-3	Satellite-4	Satellite-Mean	Bahadurabad Station
FAA_rice_	Correlation Coefficient	0.749	0.443	0.622	0.306	0.798	0.510
	Significance	0.001	0.075	0.008	0.233	0.000	0.036
FAA_tea_	Correlation Coefficient	0.740	0.308	0.714	0.141	0.747	0.487
	Significance	0.001	0.229	0.001	0.588	0.001	0.047

**Table 4 entropy-21-00722-t004:** The performances of the five models for the remote sensing series.

	BESA	CESA	MLE	OLS	YW
R	0.858	0.751	0.807	0.783	0.776
RMSE (m^3^/s)	2466	3393	2799	2990	2985
NSE	0.723	0.476	0.643	0.593	0.594

**Table 5 entropy-21-00722-t005:** The performances of the five models at the Bahadurabad station.

	BESA	CESA	MLE	OLS	YW
R	0.912	0.840	0.866	0.840	0.855
RMSE (m^3^/s)	6712	9902	7998	8812	8755
NSE	0.823	0.615	0.749	0.695	0.699

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
