# Peer review of "Application of the Entropy Spectral Method for Streamflow and Flood-Affected Area Forecasting in the Brahmaputra River Basin"

_entropy, 2019, doi:10.3390/e21080722_

Round 1

Reviewer 1 Report

The paper presents entropy spectral method  for flood-season streamflow forecasting in the basin of the Brahmaputra River. Research has a potential for practical application - developed models can be coupled with existing hydrological models to enhance the forecasting accuracy. I suggest minor improvements before publication in MDPI Entropy journal.

Line 15, 80: instead of “traditional models” provide concrete methods;

Consider to provide the research workflow scheme;

You use the name: configuration entropy spectral analysis (CESA) - isn’t it configurational entropy spectral analysis?

You missed some fundamental works:  Singh V.P., 1997, The use of entropy in hydrology and water resources,   

Shannon C.E., 1948, A mathematical theory of communication,   

Akaike H.  1973 Information Theory and an Extension of the Maximum Likelihood Principle.  

Provide the original methodology of Normalized Differences Water Index (NDWI). Discuss the spatial resolution, accuracy and errors basing on relevant literature;

Provide some numbers on land use/ land cover;

What is the spatial distribution of flood-affected areas? Alongside the whole river? I think that calculation of correlation between water flow and FAA should be based on location analyses. Provide FAA location map in relation to monitoring cross-sections;

Discuss Your results with the other BESA-CESA comparison study -  Maximum entropy spectral analysis for streamflow forecasting [pos. 13]

List of references: Cited journals should be abbreviated according to ISO 4 rules.

Author Response

Responses to Reviewer 1: Please see the attachment.

Reviewer 2 Report

This study compared monthly streamflow forecasting in the Brahmaputra River Basin in India using BESA and CESA with traditional methods, also forecasted the flood affected areas. Two sets of data were used for training purposes, remote sensing and ground observations. It is an interesting work, although it needs some more clarifications. Please see below my comments:

Major comments:

Line 78, authors mention that entropy spectral analysis methods have been rarely used in time series obtained from microwave remote sensing observation. Please mention previous works that already have used these methods and data, and their major findings in the introduction section.

Please use land use map as a background in Figure 1 to show the rice and tea harvest area in the region.

Please briefly explain the traditional methods in the method section (MLE, OLS and YW).

Line 255; Why average streamflow of four satellite sites were used for monthly forecast? Table 2 shows that Satellite 1 for rice area and Satellite 4 for tea area give better results. Why not using them instead?

Line 257; AR models and their purposes of using them here were not discussed in the methodology section and made me confused a bit here. Please clarify this.

Lines 261:263; it will be nice to see the results of this analysis in a figure or mention NSE values when training period is 15, 30, 45 moths.

Line 269; did each year include two values for dry and wet seasons? And what months of year were considered in each season?

Figures 3 & 4, what does lead time per month mean here?

Line 284; no clear overestimation pattern can be seen in Figure 5, have you checked what those outliers are?

Lines 286-287; this statement is not supported by Tables 3 & 4, these two tables show that models performances are better at the Bahadurabad station than remote sensing series.  

Section 3.4; results of logistic method was not mentioned here and the fit in Figure 6 actually is more like a logistic or non-linear regression fit rather than a segmented regression fit.

Section 3.4; what period of data was used for training? Authors mentioned that 2008-2017 was used for validation.

Some of the conclusions authors made in this work is not supported by their results and need more clarifications and better explanations to clarify it.  

Minor comments:

Lines 313 & 315; Figure 5 should be Figure 6.

Author Response

Responses to Reviewer 2: Please see the attachment.

Round 2

Reviewer 2 Report

I thank authors for providing detailed answers to my comments.

One minor comment: Figure 2. Is green color rice and tea together or just tea? Please show them separately.

Author Response

Please see the attachment - Response (Round 2)
